# Observational evidence for cloud cover enhancement over western European forests

Adriaan J. Teuling[1], Christopher M. Taylor[2,3], Jan Fokke Meirink[4], Lieke A. Melsen[1], Diego G. Miralles[5,6], Chiel C. van Heerwaarden[7], Robert Vautard[8], Annemiek I. Stegehuis[8], Gert-Jan Nabuurs[9] & Jordi Vilà-Guerau de Arellano[7]

Forests impact regional hydrology and climate directly by regulating water and heat fluxes. Indirect effects through cloud formation and precipitation can be important in facilitating continental-scale moisture recycling but are poorly understood at regional scales. In particular, the impact of temperate forest on clouds is largely unknown. Here we provide observational evidence for a strong increase in cloud cover over large forest regions in western Europe based on analysis of 10 years of 15 min resolution data from geostationary satellites. In addition, we show that widespread windthrow by cyclone Klaus in the Landes forest led to a significant decrease in local cloud cover in subsequent years. Strong cloud development along the downwind edges of larger forest areas are consistent with a forest-breeze mesoscale circulation. Our results highlight the need to include impacts on cloud formation when evaluating the water and climate services of temperate forests, in particular around densely populated areas.

[1] Hydrology and Quantitative Water Management Group, Wageningen University & Research, Droevendaalsesteeg 3a (Lumen Building), 6708PA Wageningen, The Netherlands. [2] Centre for Ecology and Hydrology, Wallingford OX10 8BB, UK. [3] National Centre for Earth Observation, Wallingford OX10 8BB, UK. [4] Royal Netherlands Meteorological Institute, 3730AE De Bilt, The Netherlands. [5] Department of Earth Sciences, VU University, 1081HV Amsterdam, The Netherlands. [6] Laboratory of Hydrology and Water Management, Ghent University, B-9000 Ghent, Belgium. [7] Meteorology and Air Quality Group, Wageningen University & Research, Wageningen 6708PA, The Netherlands. [8] LSCE/IPSL, Laboratoire CEA/CNRS/UVSQ, 91191 Gif-sur-Yvette, France. [9] Environmental Research (Alterra), Wageningen University & Research, 6708PA Wageningen, The Netherlands. Correspondence and requests for materials should be addressed to A.J.T. (email: ryan.teuling@wur.nl).

Forests are globally valued for their role in mitigating climate change through their ability to store large quantities of carbon[1–3]. Forests also have an impact on the global climate directly by altering the land surface water and energy balances[1,4]. However, at scales of 100–1,000 s km, it has been shown that forests can also alter the hydrological cycle through indirect effects on clouds and precipitation caused by land surface–atmosphere interactions[5]. This could have profound implications for forest management, spatial planning and local climate change mitigation. Both the process of cloud formation and the indirect impacts of land surface conditions on clouds are still poorly understood, in spite of its central role in the climate system and climate change projections[6]. As a result, model simulations and projections involving these processes show a wide disparity[7].

Direct (thermodynamic) effects of forests on the exchange of water and energy at the land surface are generally well understood. Forests have a lower albedo (that is, they appear darker), which causes them to absorb more of the Sun's energy than their surroundings[1,4]. Forest canopies also provide a rougher surface[8], thus promoting an efficient exchange of heat, moisture and momentum between the land surface and the atmosphere, whereas tempering the surface temperature and further increasing the net available energy. The resulting excess energy is either used for evaporation or transpiration of water (together referred to as evapotranspiration), or for heating of the well-mixed atmospheric boundary layer (ABL)[9]. Indirect biophysical effects of forests are more uncertain. Higher evapotranspiration generally promotes shallow cumulus development[10,11], although under certain atmospheric conditions low rather than high soil moisture conditions are known to enhance cloud formation and precipitation[10,12]. Other studies have also highlighted the importance of sensible heat fluxes, frictional convergence and heterogeneity-induced mesoscale circulation for cloud formation[13–15]. In addition, emissions of biogenic volatile organic compounds (BVOCs) by trees promote cloud cover[16]. The highly reactive BVOCs oxidize rapidly to form secondary organic aerosols[17], which can grow through condensation and coagulation to the size of cloud condensation nuclei.

Observational studies disagree on the direction of forest impact on clouds. In eastern Amazonia, local increases in cloud cover have been reported over deforested areas[18,19]. Contrastingly, summertime clouds were found to form over natural bushland along the bunny fence in semi-arid western Australia but not over adjacent agricultural land[20], consistent with higher evapotranspiration and albedo of forest under semi-arid conditions[21,22]. Studies in the south-eastern and mid-western United States found more clouds over forest and crop–forest boundary[23,24], suggesting a mesoscale forest-breeze circulation driven by differences in sensible heat fluxes. Contrastingly, it was noted that during summer drought in the central United States, shallow cumulus occurred more frequently over lightly vegetated than heavily vegetated landscapes[25]. For boreal forests, it was shown that their BVOC emission favours cloud formation[16]. With multiple processes affecting preferred cloud formation, a better understanding of the magnitude and direction of the effect of temperate forest on clouds and climate is needed, in particular in western Europe where many forests are located near densely populated areas.

To exclude confounding effects of topography as a static lifting mechanism that facilitates cloud formation[24], we focus our analysis on the largest temperate forests in Europe without pronounced orography (Landes and Sologne) and with sharp forest–cropland contrasts in several directions. This allows a study of regional-scale land-use effects in isolation of confounding orographic effects. Most other large forest regions in Europe are located in hilly or mountainous areas[26] where land-use effects cannot be isolated from orographic effects based on observations alone. The forest size (Landes has a forest area of over 12,000 km$^2$) is also large enough for ABL conditions to reflect forest surface conditions. Landes and Sologne experience similar climate conditions, with June–August (hereafter JJA) temperatures of 19.3 °C and 20.9 °C, and JJA precipitation of 150 and 143 mm for Sologne and Landes, respectively, although their species composition differs. In Sologne, the original and dominant tree cover of broadleaf species exists, dominated by oak (*Quercus petraea* and *Quercus robur*). The more southern Landes region is a planted maritime pine (*Pinus pinaster*) forest.

Here we analyse cloud frequency over the Landes and Sologne forest regions based on a decade of high-resolution spatial (down to 1 km) and high-frequency temporal (15 min) observations from the geostationary Meteosat Second Generation (MSG) satellite. We evaluate cloud frequency by the fraction of the time (daylight only, 6–18 UTC) that clouds are detected within a pixel. To increase the robustness of our results, we use two fully independent cloud detection algorithms (see Methods). The Cloud Physical Properties algorithm (MSG-CPP) uses information from multiple (low-resolution) MSG channels, whereas the High Resolution Visible algorithm (MSG-HRV) only uses the highest resolution visible information. Our analysis reveals a strong increase in cloud cover over large forest regions in western Europe. In addition, we show that widespread windthrow by cyclone Klaus in the Landes forest led to a significant decrease in local cloud cover in subsequent years, suggesting a long-term impact of climate extremes on forest ecosystems and land surface–atmosphere interactions. Strong cloud development along the downwind edges of larger forest areas are consistent with a forest-breeze mesoscale circulation. We conclude that forest impact on local climate conditions is more complex than previously thought and can include effects on cloud formation, mesoscale circulation and possibly the initiation of deep convection.

## Results

**Cloud frequency differences.** Both Landes and Sologne represent large forest areas surrounded by crop and grassland (Fig. 1a,b) typical for much of the western European landscape (Supplementary Fig. 1a). In contrast to other forested regions, orography is nearly absent with maximum (gradual) elevation differences in the order of 100–200 m over the region (Supplementary Fig. 1b–d). For both regions, summertime clouds are found to occur more frequently over forest than over surrounding agricultural land, with the spatial pattern of cloud frequency closely resembling forest cover (Fig. 1c–f). The fractional increase in JJA cloud frequency over forest is generally in the range of 0.05–0.15. We find similar spatial patterns for the two independent cloud retrieval products (see Methods). Sensitivity analyses on cloud cover estimates (see Methods and Supplementary Figs 2–5) show that our results are robust over a range of thresholds for cloud detection. Differences in daily JJA cloud frequency between forest and non-forest boxes are also highly significant (two-sample $t$-test, $P < 0.05$), but stronger for Landes ($P = 10^{-6}$ and $10^{-5}$ for 2004–2008) than for Sologne ($P = 0.01$ and $0.02$ over the same period). The non-forested boxes do not significantly differ ($P = 0.54$ for Landes and $P = 0.88$ for Sologne), indicating that the local forest-induced changes in cloud frequency are independent of regional-scale gradients in cloud climatology. The presence of a strong north–south gradient in cloud cover for Sologne shows that local land-use

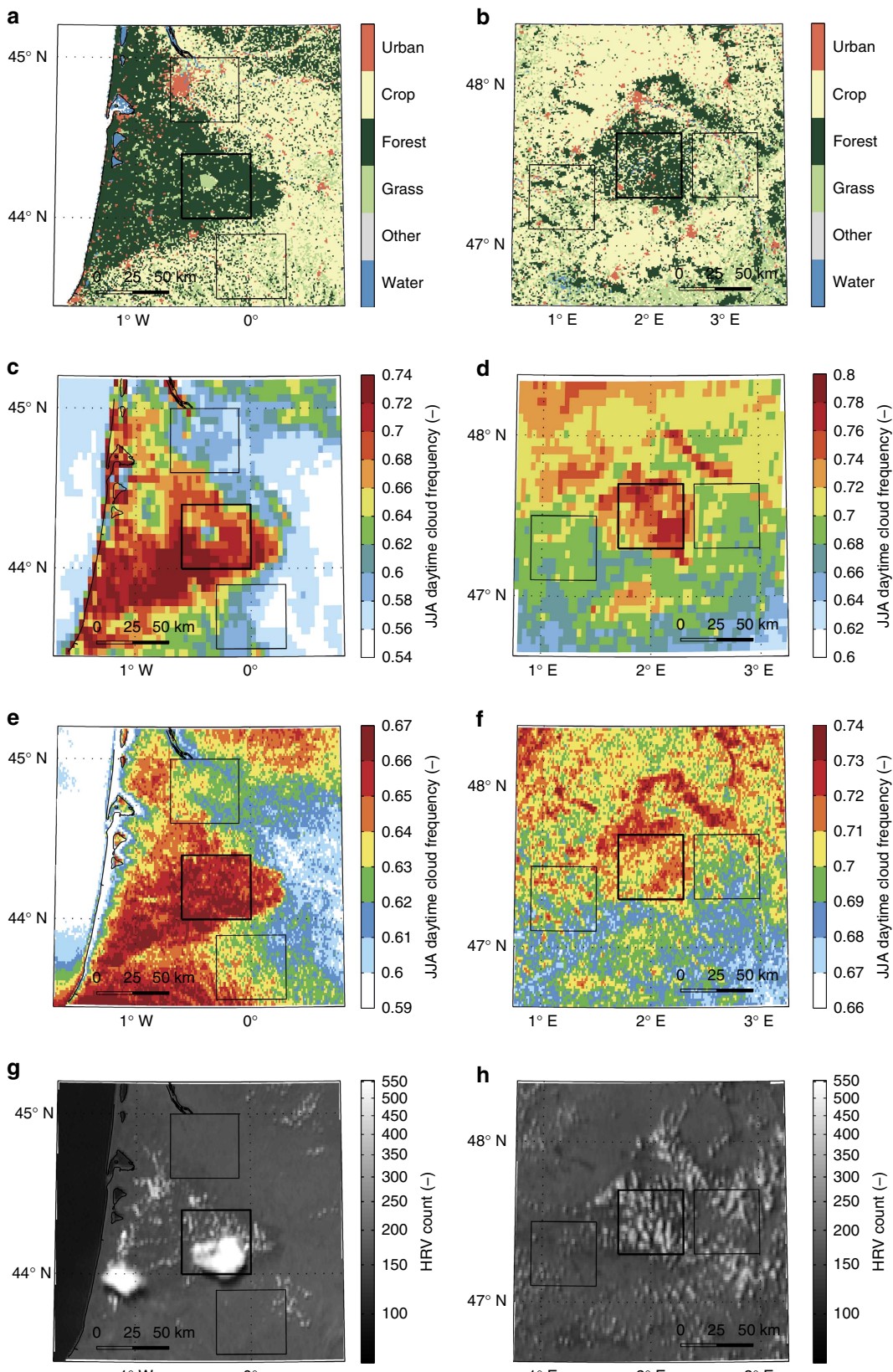

**Figure 1 | Forest cover and summer cloud occurrence. (a,b)** Land cover maps for the Landes (**a**) and Sologne (**b**) regions. (**c–f**) Mean JJA cloud frequency (2004–2008, 6–18 UTC) for the physics-based MSG-CPP product (**c,d**) and the empirical MSG-HRV product (**e,f**). (**g,h**) MSG channel 12 snapshots for 17 July 2006, 13 UTC (**g**) and 1 May 2012, 15 UTC (**h**). Black squares indicate forest (thick line) and non-forest (thin line) focus regions for visual reference and analysis in Fig. 2.

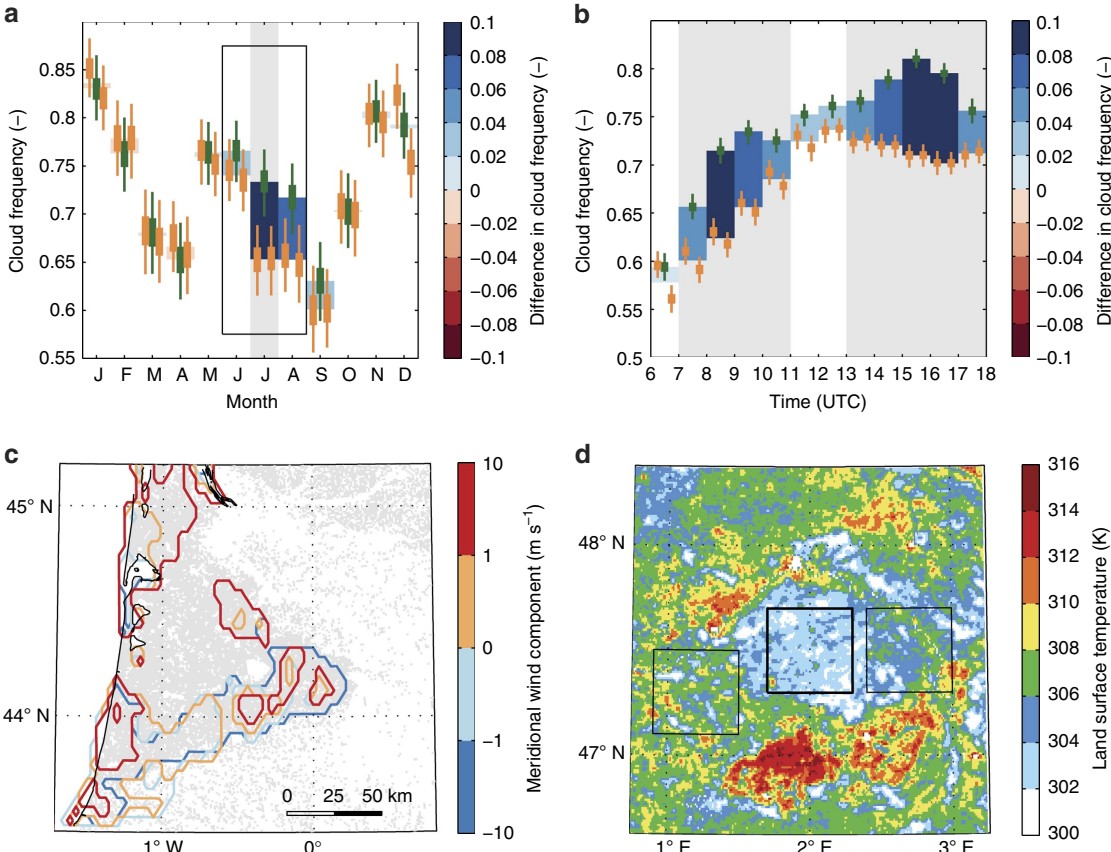

**Figure 2 | Temporal and spatial patterns of preferred cloud occurrence over forest.** Seasonal (**a**) and diurnal (**b**) evolution of mean JJA (indicated by box in **a**) difference in cloud frequency for Sologne region. (**c**) Impact of regional (10 m) meridional wind component on JJA cloud frequency for Landes. (**d**) Average July LST (2002–2014) from MODIS Aqua for Sologne. In **a,b**, thick (thin) vertical lines indicate 50 (95) percentile intervals obtained from bootstrapping for the forest (green) and non-forest boxes (orange) in Fig. 1. Shading indicates periods of significant difference at the 95% confidence level. In **c**, contours represent 90th percentiles to account for differences in mean cloud frequency with wind conditions. It is noteworthy that positive meridional wind is wind blowing from south to north. Cloud frequency is based on the MSG-CPP product. See also Supplementary Fig. 17.

effects are superimposed on larger-scale weather patterns and potentially also patterns in soil moisture availability at shorter timescales[27]. Although the climatology averages over all synoptic conditions including days with full (or absence of) cloud cover, example snapshots (Fig. 1g,h) and animations (Supplementary Movies 1–4 and Supplementary Figs 6–9) based on high-resolution imagery reveal that conditions exist in which clouds occur only over forest. In isolated cases, this can even develop into preferred triggering of deep convection (17 July 2006 case).

**Temporal dynamics**. Figure 2 explores some key temporal and spatial aspects of the contrast in cloud frequency. Figure 2a reveals the existence of a seasonal switch in cloud preference for Sologne. During JJA, there is a significant preference for clouds to occur over the forest, whereas outside of this period differences in cloud frequency are not significant. We use the Enhanced Vegetation Index (EVI) to investigate possible links with the seasonal development of vegetation in both regions. In comparison with the widely used Normalized Difference Vegetation Index, the EVI is more sensitive to biomass and canopy structure. In Sologne, EVI is initially higher over the surrounding non-forest areas (that is, forest surroundings greener than forest; Supplementary Fig. 10) but decreasing strongly during summer due to harvesting, leading to a strong surface

warming (Supplementary Fig. 11). For Landes, these seasonal dynamics are less pronounced. Here, the spatial EVI contrast is generally smaller, and both higher and lower EVI values can be found in the surrounding areas during late summer, suggesting that canopy structural differences alone are not a main determinant of the observed differences in cloud frequency. There are pronounced and significant difference in the diurnal cycles of JJA cloud cover over forest and non-forest (Fig. 2b and Supplementary Figs 12 and 13). Over forests, clouds develop earlier and faster than over the surrounding land (peak around 08:00 h UTC), while persisting longer into the evening (peak difference 15:00–16:00 h UTC). This probably reflects an earlier onset and longer duration of the thermal activity and moistening[11] of the ABL over forests. There is no significant difference between 11 and 13 UTC, probably reflecting the fact that clouds have developed fully over both forest and surrounding areas.

**Impact of synoptic conditions**. We test for the presence of forest-breeze mesoscale circulations[15] by classifying all days by (meridional) wind speed (Fig. 2c and Supplementary Fig. 14c) and wind direction (Supplementary Figs 15 and 16). This analysis reveals that the spatial pattern in cloud occurrence at the larger Landes forest is strongly impacted by wind conditions. We find a consistent pattern of higher cloud frequency at the

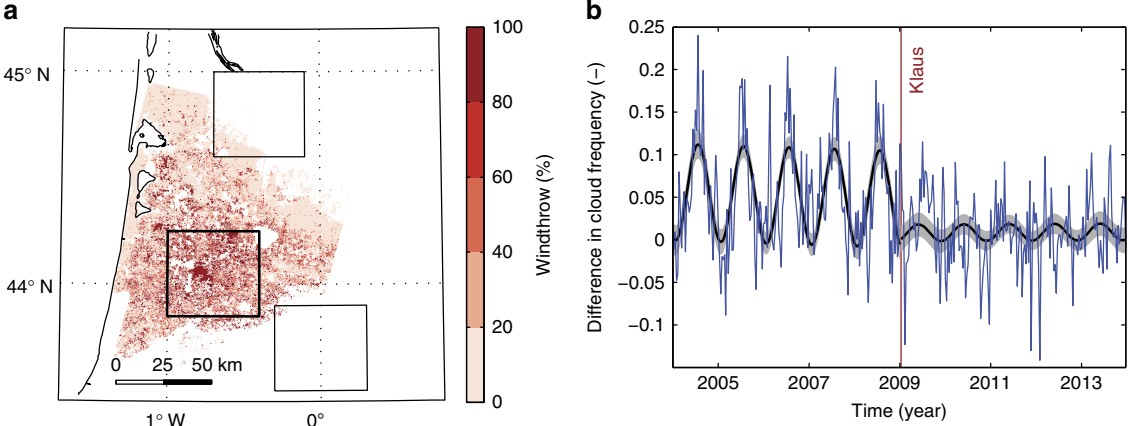

**Figure 3 | Impact of cyclone Klaus on Landes.** (**a**) Percentage of windthrow following the passage of Klaus on 24 January 2009 (data: IGN). (**b**) Changes in seasonal dynamics of differences in cloud frequency (blue line) associated with Klaus (red vertical line) based on MSG-CPP and calculated as the average over forest (thick box in **a** covering area with maximum windthrow) minus the average over non-forest areas (two thin boxes in (**a**). Ten-day average differences were fitted with a squared sine with trend (thick black line). Shading indicates 95% confidence bounds for the fit.

downwind (and to a lesser extent, cross-wind) edge of the forest over all wind directions. This edge effect is most pronounced at higher regional average wind speeds ($>1\,\text{m s}^{-1}$). These patterns are consistent with a forest-breeze circulation driven by locally enhanced heating over the forest[1,4], as previously observed for Landes[8]. In calm conditions, convergence over the forest favours cloud development. Under moderate winds, the thermally induced surface pressure gradient weakens at the upwind edge, due to advection of cool air into the forest. By contrast at the downwind edge, the local forest breeze opposes the synoptic flow, enhancing convergence and providing favourable conditions for convection. No clear relation between patterns of cloud frequency and regional wind conditions was found for the smaller Sologne forest (Supplementary Figs 14 and 16) where edge effects are more difficult to detect. We also did not find a strong dependence between synoptic conditions (measured by sea level pressure) and difference in cloud frequency (Supplementary Fig. 17), suggesting that differences in cloud cover between forest and surrounding areas occur under a range of synoptic conditions possibly involving different dominant mechanisms. The thermally induced forest breeze is likely to be strongest in June and July. Analysis of land surface temperature (LST) observations (Fig. 2d and Supplementary Fig. 11) reveals a progressive increase in LST over the surrounding areas during the summer season. The stable LST over forest probably reflects sustained evapotranspiration rates throughout the summer season, whereas the strong increase in LST over surrounding areas (up to 10 K for Sologne) is indicative for declining evaporation rates. The associated increase in thermal convection will counteract the increased thermal convection over forest due to the lower albedo.

**Impact of cyclone Klaus.** On 24 January 2009, cyclone Klaus made landfall over southern France with a core pressure reaching as low as 965 hPa and storm gusts of up to $55\,\text{m s}^{-1}$ recorded at low-level stations[13], leading to extensive damage across southern France and the northern Iberian Peninsula. The Landes forest was hit particularly severely. Many stands experienced extensive damage (Fig. 3a) by windthrow or windsnap (that is, trees uprooted or broken by wind), leading to an estimated 45 Mt of $CO_2$ emission[28]. Owing to the spatial

extent and near-instantaneous occurrence of the damage, Klaus provides a unique possibility to investigate the effects of sudden forest removal on cloud frequency. Large changes in JJA cloud frequency occurred over the region that suffered most damage (Fig. 3b and Supplementary Fig. 3). In the 5-year period, 2009–2013 following Klaus, local changes in cloud frequency up to $-0.20$ were found with respect to the period 2004–2008 ($P=0.001$ over the forest box in Fig. 3a), whereas no significant changes were found for boxes outside the storm-affected area ($P=0.42$ and $P=0.58$) nor for Sologne ($P=0.63$). The timing of the sudden change is also consistent with the passing of Klaus. Before 2009, the difference shows a marked seasonal cycle peaking in summer with an amplitude of 0.114 (0.093–0.135, 95% confidence), reducing by almost an order of magnitude to 0.020 (0.002–0.038) after Klaus. The recovery time of these forests, at least when it comes to their impact on cloud formation, is thus well over 5 years and possibly much longer.

**Discussion**
Based on the results, we provide a conceptual model for temperate forest impact on clouds (Fig. 4). Differences in the surface energy balance probably affect cloud occurrence, as several studies have shown that changes in convection due to changes in the land surface energy balance alone can create differences in cloud cover[10,11]. Our results do not contradict the important role of albedo that was hypothesized in earlier studies[21]. Albedo values for the Sologne and Landes forests are typically in the range of 0.11–0.14, whereas the surrounding areas are more reflective with albedo values in the range of 0.16–0.19 (Supplementary Fig. 18). Such differences are large enough to impact the growth and properties of the ABL. Differences in the partitioning of available energy also drive ABL dynamics and the timing of the onset of shallow cumulus formation[11], but they might be of less importance than albedo differences[9]. Nonetheless, our results are consistent with observed higher sensible heat fluxes over temperate forest[4], leading to a growing ABL and a forest breeze. We can reconcile our results with studies over Amazonia[18,19] by recognizing that in both cases higher sensible heat fluxes trigger preferred cloud formation[4,23]. Although this may seemingly contradict the cooler LSTs over forest, differences in roughness prevent LST from being a direct measure of sensible heat flux and temperature-sensitive

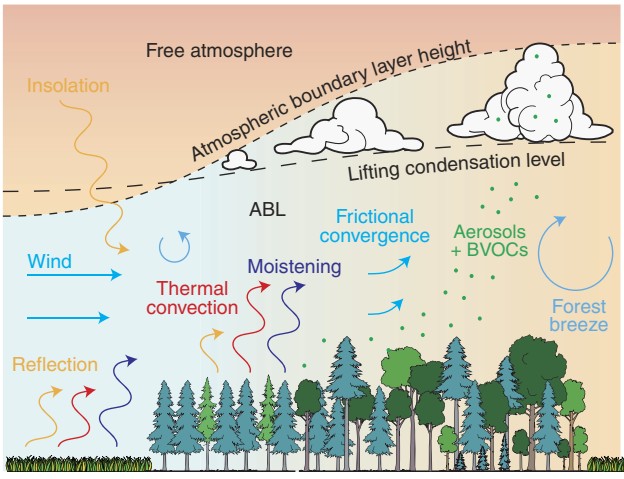

**Figure 4 | Possible pathways of temperate forest impact on convective cloud formation.** Background shading indicates potential temperate (blue = cooler, red = warmer). Arrows indicate radiative (orange), heating (red), moistening (blue), mechanical (light blue) and biogeochemical (green) processes. Over forest, less reflection of incoming solar radiation leads to larger turbulent exchange and thermal convection, increasing both the lifting condensation level and ABL height. This process is amplified by frictional convergence and the development of a forest breeze, and possibly the release of BVOCs.

flux partitioning over forest[4,9] might not be independent of cloud cover conditions. It should be noted that both a lifting mechanism (provided by sensible heat flux) and sufficient moisture (provided by latent heat flux) are required for development of convective clouds. Owing to the higher net radiation provided by the albedo effect and increased roughness, temperate forests can have higher sensible as well as latent heat fluxes[1,4].

Aerodynamic roughness has also been identified as an important factor in controlling not only the surface energy balance and mixed layer conditions[29,30], but also leading to enhanced turbulence, frictional convergence due to slowing air masses, edge effects and/or mesoscale circulation[10,14]. The strong effect of wind conditions on patterns of cloud enhancement suggest the development of mesoscale circulations[8] (forest breeze) superimposed on background wind, possibly affected by changing combinations of land surface energy balance differences and roughness. Although the existence of a forest breeze and its contribution to differences in cloud conditions has been confirmed by dedicated small-scale experiments in Landes in the framework of HAPEX-Mobilhy[8], the contribution of these effects at the regional scale remains to be tested. In addition to thermodynamic processes, the large fetch and residence time (~hours) is also favourable for the formation of secondary organic aerosols, driven by the larger BVOC emission rate of trees at the Sologne and Landes forests. Increased cloud condensation nuclei concentrations in the downwind direction could also contribute to the observed effect of wind conditions on patterns of cloud enhancement[31]. In reality, there is likely to be a complex interplay of the physical and biogeochemical processes depicted in Fig. 4, whereby the relative contribution of each process will depend on synoptic conditions, vegetation seasonality and soil moisture. Quantifying these dynamic contributions and their interplay should be a priority area for future research.

Our findings have important implications for the evaluation of water and climate services provided by temperate forests in Europe[21,32]. The increase in cloud cover has the potential to locally offset the warming effect due to lower albedo[16] and can lead to larger light and water use efficiency in forest ecosystems[21]. Our results also highlight the need to include indirect water cycle impacts when evaluating the impact of climate extremes such as cyclone Klaus on biogeochemical cycles and energy exchange[28,33]. Anecdotal evidence shows that forests can even act as a source region for deep convection, thereby possibly intensifying the hydrological cycle over land. Our study also contributes to an ongoing debate on the importance of land surface heterogeneity, either by land use or soil moisture, on clouds and precipitation[15,27,34]. Although our study focused on forests in absence of topography, we believe the mechanisms behind cloud cover enhancement will also play a role in the presence of a static lifting mechanism. However, as forest cover correlates spatially with topography[26], disentangling these effects can no longer be done based on observations alone and will require a systematic modelling approach. Future research will have to learn whether convective cloud preference over forests is also associated with triggering of deep convection and increased rainfall. Possibly, the preference for clouds to form over forest can one day be exploited, to mitigate the local impacts of a warmer climate, in particular near urban areas.

## Methods

**Data sets.** The primary data in this study are based on measurements from the Spinning Enhanced Visible and InfraRed Imager (SEVIRI) on board the MSG series of satellites. Ten years of data were used for the period 2004–2013 at a 15 min resolution, consisting of over 350,000 images in total. For most of the analysis we focused on the daytime (6–18 UTC) hours of the summer season (JJA), reducing the number of images to $10 \times 92 \times 48 = 44,160$ per region. Data of 10 m wind speed were obtained from ERA-Interim. Moderate Resolution Imaging Spectroradiometer (MODIS) albedo and EVI products were obtained from lpdaac.usgs.gov. MODIS LST data were taken from a recent global analysis of LST dynamics[35]. The storm damage map for Klaus was provided by Inventaire Forestier National (available from http://inventaire-forestier.ign.fr/cartoklaus/carto/afficherCarto, version 2009). Land-use maps for 2010 were taken from HILDA[36–38] (available from www.wageningenur.nl/hilda).

**CPP algorithm.** We used SEVIRI-based cloud products generated with the MSG-CPP algorithm at KNMI and available via http://data.knmi.nl. To avoid false detection, we only identified clouds when the cloud optical thickness exceeds 0.3. The algorithm first identifies cloudy pixels using a series of threshold and spatial coherence tests on the measured visible/near-infrared reflectances and infrared brightness temperatures[39]. In a second step, cloud optical properties (optical thickness and particle size) were retrieved by matching satellite observed reflectances at visible (0.6 μm) and near-infrared (1.6 μm) wavelengths to simulated reflectances of homogeneous clouds composed of either liquid or ice particles. The thermodynamic phase (liquid or ice) is determined as part of this procedure, using a cloud-top temperature estimate as additional input[40,41]. MSG-SEVIRI shortwave channels were calibrated with MODIS measurements[42].

**HRV algorithm.** A second empirical cloud product (MSG-HRV) was derived independently using the SEVIRI HRV broadband channel (0.4–1.1 μm). Cloud cover was identified when radiation measurements (at ~1 km pixel scale) exceeded the climatological radiation in the absence of clouds for that pixel by a constant threshold (ten counts). The clear-sky climatology was constructed by identifying the steepest section in the empirical cumulative distribution functions of all measurements over 10 years for every hour of the day and for every 10-day period in the year (400 values in total). We found that this procedure produces patterns of clear-sky radiation which agree well with MODIS albedo maps (Supplementary Fig. 18).

**Code availability.** The codes used for analysis and production of the MSG-HRV cloud product are available from the corresponding author on reasonable request.

**Data availability.** The data sets generated during and/or analysed during the current study are available from the corresponding author on reasonable request.

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

## Acknowledgements

We thank Barry Gardiner (INRA) and colleagues for their help with the Klaus storm data. D.G.M. acknowledges financial support from The Netherlands Organization for Scientific Research through grant 863.14.004. G.-J.N. acknowledges support from the H2020 PEGASUS project with grant agreement 633814.

## Author contributions

A.J.T. designed the study and carried out the analyses. J.F.M. provided the MSG-CPP data and helped initiating the study. C.M.T. provided the MSG-HRV and Aqua LST data and helped with the analysis. C.M.T., R.V., L.A.M. and D.G.M. provided additional analysis. All authors contributed to the writing and the interpretation of the results.

## Additional information

**Competing financial interests:** The authors declare no competing financial interests.

**Publisher's note**: 

