## [Peer review File · Nature Communications]

Reviewer #1 (Remarks to the Author):

This is an interesting paper on the dynamics of cloud cover over European forests and the consequences of forest cover disturbance on trends and variability. The authors show an increase in cloud cover over forested regions. Moreover, they provide evidences of a disruption of local cloud formation because of the impact of cyclone Klaus on forest cover.

In my opinion, despite the originality, consistent narrative and methods and a robust time-series of satellite data, the paper presents very localized results that are difficult to compare with more large-scale results, specially the processes occurring in Amazonia. While Amazonian forests consist mostly of a continuum permeated by deforested lands, European forests are actually embedded in an agricultural matrix, which is expected to change the regional dynamics of clouds and rainfall.

The most interesting story in the paper is related to the impact of the cyclone Klaus. I suggest the authors focus on this narrative, as the results are clear (figure 3) and the understanding of the consequences may be applicable to other punctual cases of cyclone strikes in forests. This is likely to create important feedbacks for the carbon and water cycles of the impacted regions. What is interesting is that the recovery time of these forests may be over 5 years.

I suggest a new title: Disruption o cloud cover enhancement by cyclone Klaus activity over a Western European forest.

Also in the text there are some critical information that are missing. For instance in page Pg3: What is the area of the largest forests in Europe? You state that the increase in cloud frequency is generally in the range of 0.05-0.15. This sentence is not clear. It is not clear if this increase in cloud cover refers to the long-term trend, the changes from winter to summer time or increase from agricultural lands to forests. Moreover what are the units of the values?

It is important to give information on the degrees of freedom for your statistical tests. The windthrow impact data presented in figure 3 is a secondary dataset that must be properly referenced in the text.

The use of EVI appears in the text with no explanation. Moreover it is important to give information on land cover of the regions in addition to the information on forest cover. Otherwise it is hard to understand the contrast of EVI for the two regions.

Overall, I believe this letter must be published, however because its local focus and descriptive nature of the local processes, I believe it would only reach a more specialized cohort of readers. I hope comments can help authors to improve the paper and that this study can be published soon.

Reviewer #2 (Remarks to the Author):

The methods used for the cloud detection appear reasonable and robust. In the first case, they are using standard cloud products and there is no reason to expect that these change significantly through time in such a way that would undermine the premise of the paper: meaning that as far as I can figure out neither the sensor nor the algorithms used change around the time of Klaus, which might call into question their results. The second approach is aided greatly by the use of a geostationary satellite such that measurements for the same place can be used for identifying clouds-- i.e. it takes a significant change in apparent reflectance relative to prior observations of the same location for an observation to be identified as clouds. This approach greatly reduces the problem of identifying bright land surfaces as clouds. I see no reason to call into question their results with respect to remote sensing of clouds.

Reviewer #3 (Remarks to the Author):

Teuling et al.: Observational evidence for cloud cover enhancement over Western European forests

There has been much scientific debate concerning the effects of forests, cropland, and land-cover change on climate. While there is a strong theoretical basis for such differences, and the land surface parameterizations used with climate models codify this theory, there is little empirical evidence to confirm or refute the expected differences between forest and cropland. Differences in sensible and latent heat fluxes related to surface albedo and roughness can be seen in flux tower measurements, but the effects of these differences on atmosphere boundary layer dynamics has been much harder to observe. There have been relatively few observational studies that demonstrate these effects. This paper cites those studies, and makes its own contribution to the literature.

The authors present a very nice correlation analysis and convincingly show differences in cloud cover between forest and nearby non-forest land in two regions of France. They also present a nice analysis of changes in cloud cover following large-scale loss of trees from disturbance. There are some things they could do to clarify their argument and strengthen their conclusions.

1. It would be helpful to have a map of land cover for the Sologne and Landes study regions. Within each region, the analysis relies on the contrast in cloud cover frequency between a forest sub-region and two non-forest sub-regions. Fig. 1a,b clearly shows the difference in forest cover among these sub-regions, but one would like to know more about the non-forest area (other than it has low forest cover). Is it classified as cropland, grassland, shrubland, etc? How much of this land is urban? Supplemental Fig. 1 shows only elevation within the two domains. Supplement Fig. 10 shows vegetation greenness (EVI).

2. An example of why a land cover map would be useful is that Fig. 1 shows high cloud frequency in some locations that have low forest cover. Differences between forest and non-forest cloud frequency are clearly evident in both study regions, as seen in Fig. 1c,d. However, one can also see relatively high cloud cover in the northwestern edge of the Sologne region (Fig. 1d), where there is low forest cover. Fig. 1f, with different cloud data, shows a similar pattern and also high cloud frequency in the northeastern edge. Fig. 1e shows high cloud frequency in the southwest of the Landes region that is comparable to the forest area despite much less forest cover. What is the land cover in these regions?

3. The Sologne and Landes study regions were purposely chosen because of the sharp contrast in forest and non-forest and because these regions have slight elevation difference. One wonders, however, about the uniqueness of these two areas. Do these results generalize for other locations in western Europe, or do large elevation differences or less sharp contrasts in vegetation preclude the detection of forests influences on clouds? And if so, does that imply that factors other than forests are more critical to cloud formation over much of Europe? It would be informative to have a general discussion of this. Why isn't increased cloud cover over forests seen more generally?

4. Fig.2a shows that the higher cloud frequency over forests manifests during the summer (JJA) season over the Sologne region. There is little difference between forest and non-forest in the other seasons. The authors relate this to changes in vegetation greenness (EVI) and show EVI in Supplement Fig. 10. It is necessary to show the temporal dynamics of cloud frequency and EVI for forest and non-forest in a graph akin to Fig. 2a to confirm this.

5. Fig. 3b: It is not clear what "difference in cloud frequency" means on the axis label. Also, the graph seems to be showing a pronounced annual cycle of cloud frequency (pre-disturbance), with high cloud cover during the summer. Supplement Fig. 13a also shows a strong summer increase in cloud. However, in the earlier discussion of seasonal dynamics the authors state that "for Landes,

these seasonal dynamics are less pronounced" compared with Sologne (Fig. 2a).

6. Fig. 4 presents a conceptual model of forest influences on cloud formation. This understanding is very similar to ideas proposed by others. For example, Pitman (2003; International Journal of Climatology 23:479-510) proposed similar conceptual models of albedo and roughness effects on surface climate and clouds. The discussion of Fig. 4 would benefit from a broader review of the literature.

Reviewer #4 (Remarks to the Author):

This paper uses satellite data to investigate the impact of two temperate forests in Europe on cloud cover. The authors show an enhancement in cloud frequency over the forest compared to adjacent non-forest areas, with further evidence provided by a reduction in cloud cover when one of the forests was heavily damaged after cyclone Klaus. The authors also describe the seasonal and diurnal variability of this effect, with the cloud enhancement being restricted to summer.

The paper is generally well written and organized, except for the wind results, the implications of which I find difficult to follow. The results are a worthwhile addition to the literature, given the lack of such studies in temperate regions, and the use of cyclone Klaus as a test-case of what happens when the forest is removed is novel and interesting.

Having said that, the results are not particularly surprising, and mainly confirm what has been found in previous studies. i.e. that clouds over heterogeneous surfaces form preferentially over the warmer surface type (forest in this case, while in Amazonia it is the deforested region, but the underlying mechanisms are presumably the same). There is also limited evidence presented of the mechanisms causing the results, other than demonstrating that they are consistent with past studies (other than the wind results, which are not so clear to me). Figure 4, for example, is not a summary of the findings of this paper, but a summary of mechanisms described in previous studies that this paper is consistent with. The paper does not help us understand which mechanism is most important (e.g. sensible heat flux gradient, aerosol impacts, enhanced surface friction, etc.). Therefore, while I think the results are robust and interesting, I'm not sure if the results are of sufficient high impact to justify publication here.

Major comments:

My major complaint is with figure 2c,d and supplementary figures 13c, d. Firstly, it took me a while to realize that Figure 2c,d refer to Landes (or at least I assume so), as it's not mentioned in the caption, and figure 2a, b refer to Sologne. I'm also not quite sure if the wind directions refer to where the wind is coming from, or going towards. Figure 2d seems to suggest the former. I'm not quite sure how these are evidence of forest breezes. I am assuming (but this is not spelled out at all) that the reason is that clouds are forming preferentially when the synoptic flow would oppose a hypothetical forest breeze. But clouds could also form preferentially over the forest simply due to the higher sensible and latent heat fluxes providing higher energy for convection, and then the clouds could then be advected by the synoptic flow. Maybe it is because the text here (first paragraph page 4) is very unclear and ambiguous, but I find this section quite unconvincing.

I am also slightly confused by the diurnal cycle plots (fig2b, supp fig 13b). It appears that the enhancement is significant pretty much all of the day, except for 11-13 UTC. The reasons for this are barely discussed in the text. The afternoon peak is consistent with the presence of forest breezes driven by albedo differences, as alluded by the authors. But why is there an enhancement in the early morning? I would be surprised if at this time the thermal gradients were significant. And then, why is there no statistical difference at 11-13 UTC (at which point circulations should begin to develop)? I think this potentially provides an interesting window into understanding the mechanisms behind the results which the authors don't exploit.

As a smaller point, I'm assuming the wind results for Sologne are relegated to the supplementary material because they are not significant - any thoughts as to why?

Minor comments:

page 2: "Higher evapotranspiration generally promotes shallow cumulus development..."
I'm not sure if this comment is really justified. The authors themselves give a good list of why this is often not the case. Beyond that, even when heterogeneity is ignored (which generally leads to enhanced cloud over higher sensible heat fluxes, not higher evapotranspiration), Findell and Eltahir (2003) for example describe why both wet and dry soils can both provide a cloud advantage over homogeneous surfaces, depending on the initial profile.

Page 2: "natural bushland along the bunny fence".
The reference here (11) is wrong. Also, in the study being alluded to the native vegetation had lower evapotranspiration, so the end of the sentence ("consistent with higher evapotranspiration...") is also not entirely correct. In fact the bunny fence case study is quite consistent with the Amazonian results.

Page 3, final paragraph: "whereas in early spring and late autumn..."
The results outside June and July are not significant so I don't think this is right - statistically there is no difference between forest and non-forest clouds in all other months.

Figure 2: Need to state that 2c and d are for Landes in the figure caption (if that is the case), and also state somewhere what 'wind direction' means (from or towards).

Reviewer #1 (Remarks to the Author):

This is an interesting paper on the dynamics of cloud cover over European forests and the consequences of forest cover disturbance on trends and variability. The authors show an increase in cloud cover over forested regions. Moreover, they provide evidences of a disruption of local cloud formation because of the impact of cyclone Klaus on forest cover.

In my opinion, despite the originality, consistent narrative and methods and a robust time-series of satellite data, the paper presents very localized results that are difficult to compare with more large-scale results, especially the processes occurring in Amazonia. While Amazonian forests consist mostly of a continuum permeated by deforested lands, European forests are actually embedded in an agricultural matrix, which is expected to change the regional dynamics of clouds and rainfall.

The most interesting story in the paper is related to the impact of the cyclone Klaus. I suggest the authors focus on this narrative, as the results are clear (figure 3) and the understanding of the consequences may be applicable to other punctual cases of cyclone strikes in forests. This is likely to create important feedbacks for the carbon and water cycles of the impacted regions. What is interesting is that the recovery time of these forests may be over 5 years.

I suggest a new title: Disruption of cloud cover enhancement by cyclone Klaus activity over a Western European forest.

We appreciate the suggestion for the new title, however we believe the main contribution of our paper lies in the novel observation that forests increase local cloud cover, in contrast to existing studies for the Amazon which show the opposite. We therefor prefer the original title ("Observational evidence for cloud cover enhancement over Western European forests"). We agree with the reviewer that we didn't stress the long recovery times for the Landes forest that follow from our results. The following sentence has now been added to the Results: *"The recovery time of these forests, at least when it comes to their impact on cloud formation, is thus well over 5 years and possibly much longer."*

Also in the text there are some critical information that are missing. For instance in page Pg3: What is the area of the largest forests in Europe? You state that the increase in cloud frequency is generally in the range of 0.05-0.15. This sentence is not clear. It is not clear if this increase in cloud cover refers to the long-term trend, the changes from winter to summer time or increase from agricultural lands to forests. Moreover what are the units of the values?

In order to address the comments on the wider European context, we now provide forest cover maps for Europe and its changes over the past century in the SI. These maps show the size of the Landes and Sologne forest regions in comparison to other forest regions in Europe, which are mainly located in hilly and/or mountainous terrain. This is also clarified in the main text: *"This allows a study of regional-scale land use effects in isolation from orographic effects, whereas most other large forest regions in Europe are located in hilly or mountainous areas where land use effects cannot be isolated from orographic effects based on observations alone."* The increase in cloud cover is a fractional increase over forest with respect to agricultural land. This is now also specified in the caption.

It is important to give information on the degrees of freedom for your statistical tests. The windthrow impact data presented in figure 3 is a secondary dataset that must be properly referenced in the text.

The degrees of freedom are standard for the two-sided t-test that we used. The degrees of freedom are accounted for in the p-values that are mentioned in the text. The windthrow impact data come from IFN. This was already mentioned in the Methods section (*"The storm damage map for Klaus was provided by Inventaire forestier national (IFN)"*), but the link to the relevant website has now been added.

The use of EVI appears in the text with no explanation. Moreover it is important to give information on land cover of the regions in addition to the information on forest cover. Otherwise it is hard to understand the contrast of EVI for the two regions.

The EVI is now introduced and explained in the Results section. Land cover maps are now provided in Figure 1 instead of maps of forest cover alone.

Overall, I believe this letter must be published, however because its local focus and descriptive nature of the local processes, I believe it would only reach a more specialized cohort of readers. I hope comments can help authors to improve the paper and that this study can be published soon.

We appreciate the referee's support for publication of this work.

Reviewer #2 (Remarks to the Author):

The methods used for the cloud detection appear reasonable and robust. In the first case, they are using standard cloud products and there is no reason to expect that these change significantly through time in such a way that would undermine the premise of the paper: meaning that as far as I can figure out neither the sensor nor the algorithms used change around the time of Klaus, which might call into question their results. The second approach is aided greatly by the use of a geostationary satellite such that measurements for the same place can be used for identifying clouds-- i.e. it takes a significant change in apparent reflectance relative to prior observations of the same location for an observation to be identified as clouds. This approach greatly reduces the problem of identifying bright land surfaces as clouds. I see no reason to call into question their results with respect to remote sensing of clouds.

We thank referee #2 for his/her positive assessment of our methodology. By using two independent datasets from the same platform, we believe our results are robust.

Reviewer #3 (Remarks to the Author):

There has been much scientific debate concerning the effects of forests, cropland, and land-cover change on climate. While there is a strong theoretical basis for such differences, and the land surface parameterizations used with climate models codify this theory, there is little empirical evidence to confirm or refute the expected differences between forest and cropland. Differences in sensible and latent heat fluxes related to surface albedo and roughness can be seen in flux tower measurements, but the effects of these differences on atmosphere boundary layer dynamics has been much harder to observe. There have been relatively few observational studies that demonstrate these effects. This paper cites those studies, and makes its own contribution to the literature.

We agree with the referee that observational evidence is currently lacking that is needed to bridge the gap between observed impact of forest on land surface-atmosphere exchange from flux towers and cloud and climate impacts of forests as determined by (regional) climate models.

The authors present a very nice correlation analysis and convincingly show differences in cloud cover between forest and nearby non-forest land in two regions of France. They also present a nice analysis of changes in cloud cover following large-scale loss of trees from disturbance. There are some things they could do to clarify their argument and strengthen their conclusions.

1. It would be helpful to have a map of land cover for the Sologne and Landes study regions. Within each region, the analysis relies on the contrast in cloud cover frequency between a forest sub-region and two non-forest sub-regions. Fig. 1a,b clearly shows the difference in forest cover among these sub-regions, but one would like to know more about the non-forest area (other than it has low forest cover). Is it classified as cropland, grassland, shrubland, etc? How much of this land is urban? Supplemental Fig. 1 shows only elevation within the two domains. Supplement Fig. 10 shows vegetation greenness (EVI).

We agree with the referee that this could help to reader to interpret the results. Therefore we have replaced the previous figures 1a and 1b (showing only forest cover fractions) with land cover maps showing the main land use types in the regions. The same dataset is also used in the Supplementary Information to illustrate forest cover in Europe and its changes over the past century in response to the comments by Reviewer #1.

2. An example of why a land cover map would be useful is that Fig. 1 shows high cloud frequency in some locations that have low forest cover. Differences between forest and non-forest cloud frequency are clearly evident in both study regions, as seen in Fig. 1c,d. However, one can also see relatively high cloud cover in the northwestern edge of the Sologne region (Fig. 1d), where there is low forest cover. Fig. 1f, with different cloud data, shows a similar pattern and also high cloud frequency in the northeastern edge. Fig. 1e shows high cloud frequency in the southwest of the Landes region that is comparable to the forest area despite much less forest cover. What is the land cover in these regions?

It should be noted that both regions cover a fairly large area, and climate (and cloud cover) conditions cannot be expected to remain constant over the whole region. In fact the Sologne region lies directly south of the "Loire-divide", referring to a well-known phenomenon locally that indicates generally favourable weather conditions south of the Loire. The higher average cloud cover in the northwestern corner thus reflects a large-scale gradient in cloud cover, on which the more local forest effect is superimposed. This is now made more clear by the following sentence that was added: "The presence of a strong North-South gradient in cloud cover for Sologne shows that local land use effects are superimposed on larger-scale climate patterns, and potentially also patterns in soil moisture availability at shorter timescales⁴¹."

3. The Sologne and Landes study regions were purposely chosen because of the sharp contrast in forest and non-forest and because these regions have slight elevation difference. One wonders, however, about the uniqueness of these two areas. Do these results generalize for other locations in western Europe, or do large elevation differences or less sharp contrasts in vegetation preclude the detection of forests influences on clouds? And if so, does that imply that factors other than forests are more critical to cloud formation over much of Europe? It would be informative to have a general discussion of this. Why isn't increased cloud cover over forests seen more generally?

The main reason for focussing on regions without topography in this observation-based study is that cloud patterns cannot be related to land cover alone in case forest cover is strongly correlated with topography. Unfortunately, this is the case in most of central and western Europe, where forests are mainly found on steep or elevated areas not suited for agricultural use. This is now clarified in the Introduction: "This allows a study of local land use effects in isolation of confounding orographic effects, whereas most other large forest regions in Europe are located in hilly or mountainous areas where land use effects cannot be isolated from orographic effects based on observations alone" as well as in the Discussion: "While our study focussed on forests in absence of topography, we believe the mechanisms behind cloud cover enhancement will also play a role in the presence of a static lifting mechanism. However since forest cover correlates spatially with topography⁴⁰, disentangling these effects can no longer be done based on observations alone and will require modelling."

4. Fig.2a shows that the higher cloud frequency over forests manifests during the summer (JJA) season over the Sologne region. There is little difference between forest and non-forest in the other seasons. The authors relate this to changes in vegetation greenness (EVI) and show EVI in Supplement Fig. 10. It is necessary to show the temporal dynamics of cloud frequency and EVI for forest and non-forest in a graph akin to Fig. 2a to confirm this.

While EVI plays a role in controlling the land use impact on cloud formation (through its impact on the Bowen ratio and albedo), it is by no means the only factor. Many land surface properties (albedo, LST, EVI, soil moisture, BVOC emission) that influence cloud formation co-vary through the season, and many will show a correlation with cloud occurrence. We are cautious in not overstating the relevance of such correlations in case the causality is not evident. We now show maps of LST in addition to EVI, and we have included a word of caution on the interpretation.

5. Fig. 3b: It is not clear what "difference in cloud frequency" means on the axis label. Also, the graph seems to be showing a pronounced annual cycle of cloud frequency (pre-disturbance), with high cloud cover during the summer. Supplement Fig. 13a also shows a strong summer increase in cloud. However, in the earlier discussion of seasonal dynamics the authors state that "for Landes, these seasonal dynamics are less pronounced" compared with Sologne (Fig. 2a).

We now explain in the caption how the difference was calculated: "and calculated as the average over forest (thick box in a covering area with maximum windthrow) minus the average over

non-forest areas (two thin boxes in a)". The sentence "these seasonal dynamics are less pronounced" referred to EVI and not cloud cover. This is now made more clear.

6. Fig. 4 presents a conceptual model of forest influences on cloud formation. This understanding is very similar to ideas proposed by others. For example, Pitman (2003; International Journal of Climatology 23: 479-510) proposed similar conceptual models of albedo and roughness effects on surface climate and clouds. The discussion of Fig. 4 would benefit from a broader review of the literature.

We adapted the discussion on Figure 4, also in response to comments made by other referees. In addition, we included the reference to Pitman (2003) in the discussion of roughness effects.

Reviewer #4 (Remarks to the Author):

This paper uses satellite data to investigate the impact of two temperate forests in Europe on cloud cover. The authors show an enhancement in cloud frequency over the forest compared to adjacent non-forest areas, with further evidence provided by a reduction in cloud cover when one of the forests was heavily damaged after cyclone Klaus. The authors also describe the seasonal and diurnal variability of this effect, with the cloud enhancement being restricted to summer.

The paper is generally well written and organized, except for the wind results, the implications of which I find difficult to follow. The results are a worthwhile addition to the literature, given the lack of such studies in temperate regions, and the use of cyclone Klaus as a test-case of what happens when the forest is removed is novel and interesting.

Having said that, the results are not particularly surprising, and mainly confirm what has been found in previous studies. i.e. that clouds over heterogeneous surfaces form preferentially over the warmer surface type (forest in this case, while in Amazonia it is the deforested region, but the underlying mechanisms are presumably the same). There is also limited evidence presented of the mechanisms causing the results, other than demonstrating that they are consistent with past studies (other than the wind results, which are not so clear to me). Figure 4, for example, is not a summary of the findings of this paper, but a summary of mechanisms described in previous studies that this paper is consistent with. The paper does not help us understand which mechanism is most important (e.g. sensible heat flux gradient, aerosol impacts, enhanced surface friction, etc.). Therefore, while I think the results are robust and interesting, I'm not sure if the results are of sufficient high impact to justify publication here.

We agree with the referee that the results indeed might not be "particularly surprising" for experts in the field who are familiar with all of the mechanisms involved in cloud formation and their variability across climate zones. Yet, our results showing that more clouds occur over forest are in sharp contrast to previous studies (mostly for Amazonia) showing that deforestation tends to increase cloud cover. And, contrary to what the referee claims, the new analysis of MODIS land surface temperature data (see SI) shows that forests tend to be cooler than their surroundings. We believe these findings are sufficiently novel to merit publication in Nature Communications. We believe that this work will trigger follow-up studies specially aimed at systematically unravelling the mechanisms through which forests impact clouds. This will require computationally expensive simulations with high resolution (< 1 km) atmospheric models or large-eddy simulation coupled to land and vegetation representations for specific cases, which will be different from the climate perspective followed in the current manuscript. We believe that this study (and all the cases documented) will pave the way for such numerical experiments and help them to constrain initial and boundary conditions.

Major comments:

My major complaint is with figure 2c,d and supplementary figures 13c, d. Firstly, it took me a while to realize that Figure 2c,d refer to Landes (or at least I assume so), as it's not mentioned in the caption, and figure 2a, b refer to Sologne. I'm also not quite sure if the wind directions refer to where the wind is coming from, or going towards. Figure 2d seems to suggest the former. I'm not quite sure how these are evidence of forest breezes. I am assuming (but this is not spelled out at all) that the reason is that clouds are forming preferentially when the synoptic flow would oppose a hypothetical forest breeze. But

clouds could also form preferentially over the forest simply due to the higher sensible and latent heat fluxes providing higher energy for convection, and then the clouds could then be advected by the synoptic flow. Maybe it is because the text here (first paragraph page 4) is very unclear and ambiguous, but I find this section quite unconvincing.

The wind direction and region names are now explained explicitly in the caption of Fig 2 and in the SI. The Discussion section has now been lengthened considerably to include a discussion on the "Forest-breeze" effect.

I am also slightly confused by the diurnal cycle plots (fig2b, supp fig 13b). It appears that the enhancement is significant pretty much all of the day, except for 11-13 UTC. The reasons for this are barely discussed in the text. The afternoon peak is consistent with the presence of forest breezes driven by albedo differences, as alluded to by the authors. But why is there an enhancement in the early morning? I would be surprised if at this time the thermal gradients were significant. And then, why is there no statistical difference at 11-13 UTC (at which point circulations should begin to develop)? I think this potentially provides an interesting window into understanding the mechanisms behind the results which the authors don't exploit.

The morning enhancement is likely caused by an earlier onset of thermal activity and moistening of the atmospheric boundary layer over the darker forest areas. This is also consistent with the work by Gentine et al., who show that the timing of onset of shallow cumulus provides important information on surface fluxes. This paper is now cited, and we have added the following sentence: "*This likely reflects an earlier onset and longer duration of the thermal activity and moistening of the ABL over forests*". We also agree with the referee that the manuscript needs a more thorough discussion on the underlying mechanisms. This is now provided in the form of an extended section in the Discussion. The time window 11-13 h is now discussed as well: "*There is no significant difference between 11 and 13 UTC, likely reflecting the fact that clouds have developed fully over both forest and surrounding areas.*"

As a smaller point, I'm assuming the wind results for Sologne are relegated to the supplementary material because they are not significant - any thoughts as to why?

Edge effects at Sologne are more difficult to detect because of its smaller size compared to Landes. This is now mentioned: "*No clear relation between patterns of cloud frequency and regional wind conditions was found for the smaller Sologne forest (Supplementary Figs. 14 and 16) where edge effects are more difficult to detect.*"

Minor comments:

page 2: "Higher evapotranspiration generally promotes shallow cumulus development..."

I'm not sure if this comment is really justified. The authors themselves give a good list of why this is often not the case. Beyond that, even when heterogeneity is ignored (which generally leads to enhanced cloud over higher sensible heat fluxes, not higher evapotranspiration), Findell and Eltahir (2003) for example describe why both wet and dry soils can both provide a cloud advantage over homogeneous surfaces, depending on the initial profile.

Indeed both wet and dry soils can enhance cloud cover. This is now mentioned, along with a reference to the Findell & Eltahir paper.

Page 2: "natural bushland along the bunny fence".

The reference here (11) is wrong. Also, in the study being alluded to the native vegetation had lower evapotranspiration, so the end of the sentence ("consistent with higher evapotranspiration...") is also not entirely correct. In fact the bunny fence case study is quite consistent with the Amazonian results.

We agree with the referee on the incorrect reference (should be ref. 8, now corrected) but disagree with the referee on the use of the reference. In fact, the Ray et al paper (ref. 8) states that:

"Latent heat fluxes are higher over native vegetation than over agricultural areas during summer, while sensible heat fluxes are lower. Cumulus clouds occur with higher frequency and have higher optical thicknesses, cloud liquid water contents, and effective radii over agricultural areas during the winter and over native perennial vegetation during the dry summer. This is due to higher latent heat fluxes and available energy over agriculture during winter and over native vegetation during summer. We conclude that land use differences result in differences in available soil moisture and surface energy fluxes, which in turn lead to the observed preferential enhancement of cumulus cloudiness and cumulus cloud properties."

So for the summertime conditions we refer to, clouds occur more frequent over bushland consistent with the higher latent heat flux. This is in contrast to the situation in Amazonia.

Page 3, final paragraph: "whereas in early spring and late autumn...."

The results outside June and July are not significant so I don't think this is right - statistically there is no difference between forest and non-forest clouds in all other months.

This is correct. We have added statements on the significance of these signals in the text.

Figure 2: Need to state that 2c and d are for Landes in the figure caption (if that is the case), and also state somewhere what 'wind direction' means (from or towards).

Landes was already mentioned in the caption ("*Impact of regional (10 m) meridional wind component on JJA cloud frequency for Landes.*"). We clarified the (commonly used) definition of meridional wind in the caption: "*Note that positive meridional wind is wind blowing from south to north*". Note that Figure 2d has been changed and now shows the surface temperature for Sologne (as mentioned in the caption).

Reviewer #1 (Remarks to the Author):

The authors have successfully tackled all key points raised in my comments. The paper is now solid and the results and conclusions are ready to be evaluated by the wide research community.

Reviewer #3 (Remarks to the Author):

The authors have satisfactorily addressed my comments and I am satisfied with the manuscript.

Reviewer #4 (Remarks to the Author):

I am overall satisfied with the responses. I appreciate the attempt to discuss the mechanisms in some more depth. This is (necessarily) somewhat speculative, which is fine, but I think that stating that figure 4 is 'based on the results' is a bit of an overstatement; it's primarily a list of potential mechanisms based on the literature, as correctly stated in the figure caption. A couple of in text clarifications should be sufficient to address the minor points below.

There is one aspect which is still unclear to me regarding the mechanisms. The authors state in the response that "the new analysis of MODIS land surface temperature data (see SI) shows that forests tend to be cooler than their surroundings", thus setting these results apart from studies over the Amazon. However, in the discussion they state that the "results are consistent with observed higher sensible heat fluxes over temperate forest, leading to a growing ABL and a "forest-breeze", and that "We can reconcile our results with studies over Amazonia by recognizing that in both cases higher sensible heat fluxes trigger preferred cloud formation". The latter sentence summarizes my original point that mechanistically these results are probably consistent with the ones over the Amazon (but with higher sensible heating over the forested, instead of deforested, areas). How do you reconcile the LST results showing cooler forest, with higher sensible heating over the forest, and forest breezes converging over it (as implied in figure 4)?

A final, minor, point about the discussion. When the authors state that "Differences in the partitioning of available energy...are likely of less importance than albedo differences" are they just referring to their results, or more generally in the literature? The presence of references backing up these statements suggest the latter, but I would have thought this is not always the case. In particular my understanding is that, if anything, the Amazonian results suggest the opposite.

Reviewer #1 (Remarks to the Author):

The authors have successfully tackled all key points raised in my comments. The paper is now solid and the results and conclusions are ready to be evaluated by the wide research community.

We thank the reviewer for the positive evaluation of our work.

Reviewer #3 (Remarks to the Author):

The authors have satisfactorily addressed my comments and I am satisfied with the manuscript.

We thank the reviewer for the positive evaluation of our work.

Reviewer #4 (Remarks to the Author):

I am overall satisfied with the responses. I appreciate the attempt to discuss the mechanisms in some more depth. This is (necessarily) somewhat speculative, which is fine, but I think that stating that figure 4 is 'based on the results' is a bit of an overstatement; it's primarily a list of potential mechanisms based on the literature, as correctly stated in the figure caption. A couple of in text clarifications should be sufficient to address the minor points below.

We thank the reviewer for the positive evaluation of our work. We have made some small textual changes in response to the issues raised below.

There is one aspect which is still unclear to me regarding the mechanisms. The authors state in the response that "the new analysis of MODIS land surface temperature data (see SI) shows that forests tend to be cooler than their surroundings", thus setting these results apart from studies over the Amazon. However, in the discussion they state that the "results are consistent with observed higher sensible heat fluxes over temperate forest, leading to a growing ABL and a "forest-breeze", and that "We can reconcile our results with studies over Amazonia by recognizing that in both cases higher sensible heat fluxes trigger preferred cloud formation". The latter sentence summarizes my original point that mechanistically these results are probably consistent with the ones over the Amazon (but with higher sensible heating over the forested, instead of deforested, areas). How do you reconcile the LST results showing cooler forest, with higher sensible heating over the forest, and forest breezes converging over it (as implied in figure 4)?

Indeed these statements might seem contradictory, but they are not necessarily so. To clarify the difficulties in the interpretation of LST, we added the following sentence: "While this may seemingly contradict the cooler LSTs over forest, differences in roughness prevent LST from being a direct measure of sensible heat flux, and temperature-sensitive flux partitioning over forest^{4,9} might not be independent of cloud cover conditions."

A final, minor, point about the discussion. When the authors state that "Differences in the partitioning of available energy...are likely of less importance than albedo differences" are they just referring to their results, or more generally in the literature? The presence of references backing up these statements suggest the latter, but I would have thought this is not always the case. In particular my understanding is that, if anything, the Amazonian results suggest the opposite.

We changed "are likely of less importance" into "might be of less importance" to highlight the fact there is still uncertainty on the role of albedo vs flux contributions.